# Cellulose Nanocrystal-Based Emulsion of Thyme Essential Oil: Preparation and Characterisation as Sustainable Crop Protection Tool

**DOI:** 10.3390/molecules28237884

**Published:** 2023-11-30

**Authors:** Francesca Baldassarre, Daniele Schiavi, Veronica Di Lorenzo, Francesca Biondo, Viviana Vergaro, Gianpiero Colangelo, Giorgio Mariano Balestra, Giuseppe Ciccarella

**Affiliations:** 1Department of Biological and Environmental Sciences, UdR INSTM of Lecce University of Salento, Via Monteroni, 73100 Lecce, Italy; francesca.biondo@unisalento.it (F.B.); viviana.vergaro@unisalento.it (V.V.); 2Institute of Nanotechnology, CNR NANOTEC, Consiglio Nazionale delle Ricerche, Via Monteroni, 73100 Lecce, Italy; 3Department of Agriculture and Forest Sciences (DAFNE), University of Tuscia, Via S. Camillo de Lellis, snc, 01100 Viterbo, Italy; schiavi@unitus.it (D.S.); veronica.dilorenzo@unitus.it (V.D.L.); balestra@unitus.it (G.M.B.); 4Department of Engineering for Innovation, University of Salento, Via Monteroni, 73100 Lecce, Italy; gianpiero.colangelo@unisalento.it

**Keywords:** cellulose nanocrystals, thyme essential oil, nanoemulsions, phenols, biopesticides, olive knot disease

## Abstract

Essential oil-based pesticides, which contain antimicrobial and antioxidant molecules, have potential for use in sustainable agriculture. However, these compounds have limitations such as volatility, poor water solubility, and phytotoxicity. Nanoencapsulation, through processes like micro- and nanoemulsions, can enhance the stability and bioactivity of essential oils. In this study, thyme essential oil from supercritical carbon dioxide extraction was selected as a sustainable antimicrobial tool and nanoencapsulated in an oil-in-water emulsion system. The investigated protocol provided high-speed homogenisation in the presence of cellulose nanocrystals as stabilisers and calcium chloride as an ionic crosslinking agent. Thyme essential oil was characterised via GC-MS and UV-vis analysis, indicating rich content in phenols. The cellulose nanocrystal/essential oil ratio and calcium chloride concentration were varied to tune the nanoemulsions’ physical–chemical stability, which was investigated via UV-vis, direct observation, dynamic light scattering, and Turbiscan analysis. Transmission electron microscopy confirmed the nanosized droplet formation. The nanoemulsion resulting from the addition of crosslinked nanocrystals was very stable over time at room temperature. It was evaluated for the first time on *Pseudomonas savastanoi pv. savastanoi*, the causal agent of olive knot disease. In vitro tests showed a synergistic effect of the formulation components, and in vivo tests on olive seedlings demonstrated reduced bacterial colonies without any phytotoxic effect. These findings suggest that crosslinked cellulose nanocrystal emulsions can enhance the stability and bioactivity of thyme essential oil, providing a new tool for crop protection.

## 1. Introduction

The massive use of synthetic antimicrobial agrochemicals in agrifood raises several concerns about environmental and toxicological pollution [1,2]. The synthetic pesticide market is in decline, while biopesticides, which have been estimated to conspicuously grow by 2025, could offer an innovative way to achieve greater sustainability in agricultural practices, even matching the growing preference of consumers for natural and safer products [3,4,5]. In this context, essential oils (EOs) represent a valid and ecofriendly alternative to chemicals and actual biopesticides [6,7]. These plant extracts are rich in aromatic secondary metabolites with antimicrobial effects and plant defence mechanisms against several pathogenic organisms. Phenolic compounds such as carvacrol, thymol, coumarins, limonene, eugenol, and menthol possess antibacterial, antifungal, antiviral, antimycotic, antiparasitic, insecticidal, antioxidant, and antiseptic properties. Therefore, the use of EOs and their constituents ranges from medicine to agriculture as biopesticides, and use in food preservatives and additives in food packaging materials [8]. EOs are ecofriendly products because of their volatility, which allows for their removal from the environment. However, this volatility is a limit to their bioactivity, which is also compromised by decomposition under light and oxygen exposure during application or storage. 

The application of nanotechnology leads to important innovations in products, including EO formulations, refining their physicochemical properties and functions [9,10,11,12]. EOs need to be encapsulated in emulsion form to be dispersible in water and to improve stability and bioavailability. Colloidal stability is also required to avoid gravity-related problems such as creaming or sedimentation. Nanoemulsions (NEs) offer improved wettability, improved diffusion capacity, and superior mechanical/chemical stability through the formation of nanometric droplets. These properties also reduce volatilisation and degradation of active ingredients, increasing target site penetration and bioactivity [13]. NE formulations are crucial for the water dispersion of lipophilic and/or poorly water-soluble compounds such as EOs, pesticides, or foodstuffs [14,15,16]. The following Figure 1 presents the potential features of EO NEs for agricultural application. 

These systems are generally composed of a water and an oil phase with the presence of surfactants, amphiphilic molecules which provide a decrease in the interfacial tension between the two immiscible phases. NEs differ from microemulsions (MEs) not only due to the formation of nanosized droplets, ranging from 20 to 200 nm, but also due to their kinetic and long-term stability due to their high- and low-energy methods of preparation. Furthermore, NEs require a lower quantity of surfactants than MEs, reducing potential toxicological problems [17]. The use of surfactants and emulsifiers minimises or prevents the occurrence of Ostwald ripening, prevents coalescence, and improves the antimicrobial activity of EOs [18]. Emulsifiers are a subcategory of surfactants which prevent oil droplet aggregation, increasing colloidal stability. Food-grade synthetic and natural emulsifiers are being used in NE formulations, such as polyoxyethylene sorbitan, phospholipids like lecithin, proteins, polysaccharides (gum arabic and pectin), and saponins [19,20,21]. NE stabilisation can also be carried out via solid particle addition, referred to as the Pickering method. Organic and inorganic particles retard oil evaporation and avoid colloid coalescence, improving essential oil NEs’ antimicrobial activity for cosmetic, pharmaceutic, and agrifood applications [22]. Several spherical, rod-like, and plate-like particles can be used to obtain Pickering emulsions. Spherical inorganic particles, such as titanium oxides, alumina, and silica, offer good stability but present environmental problems [23,24]. Protein nanoparticles, soy, gelatin, and other polysaccharide nanoparticles as well as polysaccharide/protein complex nanoparticles have been studied [25]. 

Cellulose nanoparticles are great stabilisers compared with other types of particles due to their biocompatibility, biodegradability, thermomechanical behaviour, and costs. Cellulose nanocrystals (CNCs), cellulose nanofibers (CNFs), and microfibrillated cellulose (MFC) have been used in NEs and MEs, in both native and modified forms [26,27,28]. Furthermore, CNCs are very interesting additives due to the better control of their morphology and chemical changes, and for the reproducibility of their emulsion formation. The hydroxyl groups on the surface of the CNCs give this nanomaterial an amphiphilic characteristic that is useful for stabilising oil–water emulsions. Indeed, the negative charge of the CNCs favours the colloidal stabilisation of NE nanoparticles [29]. CNCs display interesting biological properties when it comes to microorganism interactions, which have been recently explored to develop crop protection tools, together with other natural active ingredients, such as chitosan, starch, and gallic acid [30,31]. Furthermore, several works have reported on the smart combination of CNCs as biostabilisers and EOs as green pesticides [32,33]. CNCs-stabilised emulsions of clove, oregano, and thyme white essential oils showed enhanced antimicrobial activity against food-related microorganisms and larvae, stronger than that of pure EOs [29,33,34]. In addition, the encapsulation of thymol and eugenol EOs has been effectively achieved with a new emulsification method using an unmodified cellulose shell, achieving high antimould activity, which is useful for ecological pre- and postharvest pathogen control [35]. Thymol, the primary phenolic component of thyme (*Thymus vulgaris*) essential oil, displays rapid activity against gram-negative and gram-positive bacteria compared with other plant metabolites [36,37]. Recently, we found a great inhibition activity towards *Xylella fastidiosa* for thymol, which was improved through the carrier action of CaCO_3_ nanocrystals [38,39]. 

In this work, thyme EO (Th-EO) from an ecofriendly extraction process was characterised with GC-MS and UV-vis analysis and used for the first time as a crop protection tool against *Pseudomonas savastanoi pv. Savastanoi* (Psav). This gram-negative bacterium causes olive knot disease, and its management depends on agronomic practices and preventive chemical treatments, including with copper salts [40]. Copper-based agrochemicals can accumulate in the soil and cause harm to plants, nontarget organisms, and humans [41]. The European Commission has strict regulations on the use of such chemicals, prompting the need for alternative options. Despite some progress, olive knot disease continues to be a threat to olive crops in the Mediterranean region. Therefore, it is important to explore alternative management methods [42,43,44,45]. 

In detail, NEs were prepared by exploiting the stabilisation action of CNCs and using CaCl_2_ as an ionic crosslinking agent, resulting in the obtainment of CNCs@Th-EO NEs. Stability was investigated through a visual method, dynamic light scattering, and Turbiscan analysis. Nanodroplet morphology was observed using transmission electronic microscopy. Samples were characterised using UV-vis spectrometry to obtain absorption spectra and total phenolic content. Stability studies confirmed a long-term physical–chemical stability after 30 days of storage at room temperature (RT). Three in vitro tests showed a great inhibition action of Th-EO towards Psav that was improved with NE application. This action was confirmed via in vivo testing, which provided a reduction in epiphytic survival in infected olive plants after 7–14 days post-inoculum (dpi), without phytotoxic effects. Our data demonstrate that Th-EO is a valid biopesticide and its bioactivity is improved by emulsification in the presence of CNCs and CaCl_2_, thanks to the formation of a very stable nanodroplet suspension. 

## 2. Results and Discussion

### 2.1. Chemical Composition of Thyme Essential Oil 

Th-EO from Licofarma s.r.l. company was extracted from *Thymus vulgaris* L. using a supercritical CO_2_ extraction process. The yield of Th-EO was over 2% (*w*/*w*) and its density was 0.95 g/mL. The obtained pure oil was stored at RT until utilisation for chemical composition and further studies. Th-EO was light yellow in colour and had a characteristic, sharp odour. Extraction methods, as well as geographical location, plant species, harvest time, and climate, can influence the quality and composition of EO [46]. 

We first quantified the total phenol content (TPC) with the Folin–Ciocalteu colorimetric method (Section 3.4.2), detecting a TPC of 219 ± 19 µg GAE/mg Th-EO. The phenolic content of plant extracts significantly affects their antioxidant and antimicrobial activities [8,47]. Phenols and phenolic acid are bioactive phytochemicals consisting of a single substituted phenolic ring, and their antimicrobial activity is due to the position and number of hydroxyl groups. Toxicological mechanisms consist of membrane disruption, cell wall complexation, adhesin binding, and enzyme inactivation [48]. 

GC-MS analysis was exploited to identify the main compounds of EO. The most abundant compound in our Th-EO was *o*-cymene, followed by thymol. Other compounds were present in undetectable traces. The chemical composition obtained using GC-MS is shown in Table 1 with the chemical formulas of the identified compounds. 

Our results are partially in accordance with those found by other authors. Thyme EO is generally composed of *p*-cymene, linalool, thymol, and carvacrol. These last compounds are responsible for antimicrobial activity [49]. Some variations have also been found for thyme white essential oil, which is mainly composed of thymol and *p*-cymene [50]. 

Thymol (2-isopropyl-5-methylphenol) is the main monoterpene phenol found in extracts from plants of the *Lamiaceae* family, such as *Thymus*, *Thymbra*, and *Origanum*, but it is also present in other species. It is a very interesting substance with many potential applications in different fields, from pharmaceutical use to foodstuffs, for which it is registered by the European Commission as a safe flavouring. Nevertheless, there are many works assessing its antibacterial and antifungal activity towards a broad spectrum of pathogens [36]. 

### 2.2. Physical–Chemical Properties of Nanoemulsions 

#### 2.2.1. Preparation of CNCs@Th-EO NEs and Oil Entrapment Efficiency

The potential of EOs as biopesticides is related to nanoencapsulation strategies. In this context, the delivery of active ingredients used as pesticides via NEs is central, as shown in the recent literature [7]. 

We formulated oil in water (O/W) NEs using Tween 80 as a surfactant, ethanol as a cosurfactant to increase the solubilisation of active ingredients, and CNCs as stabilisation agents, just as described in the Materials and Methods Section 3.3. Tween 80 and Tween 20 are the most used surfactants for EO emulsions because they are effective for the achievement of colloidal stability. Furthermore, nonionic surfactants are usually applicated in agriculture for their nontoxicity, for their ecocompatibility, and because they are not affected by pH and ionic strength [16,51,52]. 

The oil/surfactant ratio (*v*:*v*) was maintained at 2:1, just as described in the Materials and Methods Section 3.3; instead, two Th-EO:CNCs ratios (*w*:*w*) were investigated to obtain NEs with different oil content (% *v/v*). CNCs were added as solid particles to form steric protection at the O/W interface with the aim of long-term stability, which is widely discussed in subsequent paragraphs. Furthermore, CaCl_2_ was investigated as an ionic crosslinker. The aim was to enhance the emulsification action of CNCs by establishing salt bridges inside the cellulose network, as has been experimented for different polysaccharide-based systems [53,54,55]. Ca^2+^ ions intercalate between cellulose nanocrystals, forming electrostatic interactions with the negatively charged carboxyl groups (–COO^−^) of nanocellulose chains. Formulations with and without the salt were prepared to evaluate the chemical composition and stability of CNCs@Th-EO NEs. The following Figure 2 presents the flowchart of the investigated procedures and methodologies. 

Table 2 presents the formulated NEs, indicating their initial composition, their oil entrapment efficiency according to UV-vis measurements, and their TPC data after preparation. 

All samples after preparation appeared homogeneously white in colour, with no visible free oil or creaming, showing the successful emulsification of Th-EO. The oil entrapment was 100%, as indicated by both the oil retention percentage data and the TPC assay. The NE TPC data are close to those of free oil at the same Th-EO concentration (see previous paragraph). 

The oil entrapment efficiency data indicated that emulsion formulation (initial oil content and CaCl_2_ addition) did not influence the Th-EO loading capacity. It can be assumed that the stabilising action of CNCs is effective enough to trap any amount of oil introduced, regardless of the EO concentration and crosslinking step. This finding is in line with previous works about cellulose-based emulsions of EOs; at low cellulose particle concentrations, the EO type (different polarity) mainly influenced the emulsion process [32]. 

Furthermore, the UV spectra of free Th-EO and NE showed no changes in absorption peak wavelengths after encapsulation, suggesting there were no structural modifications of phytochemicals following the emulsification process (see Figure 1). 

#### 2.2.2. Nanoemulsion Stability

Oil entrapment efficiency provides a qualitative assessment of the performance of a given encapsulation process, while the emulsion stability indexes relate to the ability of a given emulsifier/oil system to contrast the spontaneous phase separation. 

The four formulations were assessed daily through visual observation after solvent evaporation (time 0) and during the two weeks after preparation (at 1, 7, and 14 days). The solvent evaporation was conducted via continuous stirring (300 rpm) overnight at room temperature under the fume hood. The NEs were stored at RT and at 4 °C. The gravitational stability was verified via centrifugation of sample aliquots as described in the Materials and Methods section. Stability indexes were calculated following Equations (2) and (3). Oil retention was also analysed after 30 days by quantifying Th-EO in the remaining emulsion volume through UV absorption using Equation (1). 

First of all, surfactant addition is necessary to better solubilise EOs. NEs without Tween 80 showed complete phase separation for both oil concentrations (10 and 1.5% *v*/*v*). The incorporation of almost 5% of surfactant is generally reported for efficient nanoemulsion production due to its influence on oil droplet aggregation. Several studies have investigated surfactant flexibility and ability to tune the hydrophilic–lipophilic balance (HLB), including the synergistic effect from a mixture of nonionic surfactants [16,56]. In our work, Tween 80 was used in combination with CNCs, ensuring a loading efficiency of 100% Th-EO (see Table 2). In terms of emulsion formation and stability, a drastic creaming phenomenon was observed, reaching an SI % of 10 already after 1 day without CNCs. The role of the CaCl_2_ crosslinker and the influence of the EO concentration were then studied. The following Table 3 presents the stability parameters of CNCs@Th-EO NEs (see Table 2) stored at RT in the dark.

All NEs after solvent evaporation (time 0 d) remained homogeneous for the first 24 h, as indicated by the SI percentages. Stability towards centrifugation shear indicated a loss of stability for the 10% *v*/*v* oil sample, without a crosslinker (CNCs@Th-EO NE_2). Moreover, no considerable changes were observed for the other emulsions following centrifugation. A minor phase separation was observed for samples without CaCl_2_ after the first day of storage; SI percentages were 85% and 90%, respectively, for CNCs@Th-EO NE_2 and CNCs@Th-EO NE_4 (10% *v*/*v* and 1.5% *v*/*v* oil samples). In these samples, we observed a clear cream layer which remained unchanged over time, as indicated by the measured SI percentages (two weeks after preparation). In addition, the CNCs@Th-EO NE_2 sample showed a significant loss of oil retention (50%) in the remaining emulsion layer, as determined using oil quantification. The UV-vis spectra of the oil in the cream and emulsion layers of the CNCs@Th-EO NE_2 formulation are shown in Appendix A, which illustrates the reduction in EO retention. We observed a small decrease in emulsion volume for CNCs@Th-EO NE_2 after storage, so the loss of oil retention is not due to evaporation of the unstabilised EO but is related to the phase separation caused by the creaming phenomenon. The oil retention data were also confirmed via TPC quantification. No differences were detected or measured for NEs stored at 4 °C, demonstrating that storage temperature does not affect the produced Th-EO formulations, as shown in Appendix A.

The stability data suggested that the stabilising effect of CNCs was strong and fundamental; the ES and SI percentages were high for all samples and remained stable for 14 days. However, the best results were obtained with CaCl_2_, which enhanced the ability of CNCs to form stable emulsions due to the strong electrostatic network. The crosslinked network provided inhibition of coalescence and creaming phenomena with maximum oil entrainment, regardless of the initial EO content. This efficient EO coverage easily led to emulsion stability over time and after centrifugation, resulting in uniform droplet distribution, as confirmed via subsequent analysis. This mechanism is necessary to prevent destabilisation, as demonstrated for EO emulsions based on CNCs and CNFs [57]. 

Increasing the oil concentration in a formulation could affect the resulting nanoemulsion’s stability [58]. Despite the increase in oil content, our CNCs@Th-EO NE_1 formulation (10% *v*/*v* concentration of Th-EO) showed excellent physicochemical stability. No differences were found between CNCs@Th-EO NE_1 and CNCs@Th-EO NE_3, which have different oil contents but the same concentrations of CNCs and CaCl_2_ (1.5% *w*/*v* and 3 mM, respectively), suggesting that the increase in CNCs concentration is not necessary exploiting the ionic crosslinking of Ca^2+^. Mikulcová et al. demonstrated an improvement in the stability of EO emulsions by increasing the amount of cellulose, particularly for MFC, which provided a stronger three-dimensional fibril–droplet network than that formed by CNCs [32]. An effect of oil type, viscosity, and polarity has also been suggested [57]. Shin et al. observed a floating creamy layer for the samples with lower content of CNCs (as mg CNCs/mL of oil) that was not sufficient for the efficient EO droplets covering [50]. We suggest a synergistic action among crosslinked CNCs in the aqueous phase and the Th-EO/Tween 80 mixture of the oil fraction, which achieved oil droplet covering and coalescence blocking. 

Nanoemulsion instability can result from particle aggregation or dissolution due to the Ostwald ripening phenomenon which usually occurs on the first days after the preparation and is dependent on the oil phase fraction in the NE system [16]. Therefore, dynamic light scattering (DLS) analysis was applied to determine the size and ζ-potential of NE droplets and their colloidal stability over time, in terms of polydispersity index (PdI) and average hydrodynamic diameter (Z-average diameter). We again analysed the contribution of the oil fraction in the presence of CNCs and with and without crosslinking.

#### 2.2.3. Nanodroplet Size and Colloidal Stability over Time 

Size and ζ-potential of droplets are important parameters impacting the NEs’ colloidal stability over time. Furthermore, the PdI gives information about sample uniformity that is crucial to detect potential coalescence and/or flocculation phenomena. These data were recorded using DLS analysis, a common technique for determining the size distribution of nanocolloid suspensions. These measurements require dilution of the samples to avoid multiple scattering due to aggregation as a result of electrostatic interactions. Therefore, this light scattering assay is also suitable for determining the stability of NEs after dilution, which can affect emulsion performance [59]. 

Figure 2 reports the DLS plots, Z-average diameters, and PdI versus ageing time (days) for all the produced NEs. 

First, the four NEs showed a PdI below 0.4 and remained below 0.4 after storage, suggesting monodisperse oil droplet formation and good formulation homogeneity, consistent with agricultural applications [16]. The addition of CaCl_2_ improved colloidal stability just as observed in previous studies (paragraph 2.2.2). The formulations with CaCl_2_ (CNCs@Th-EO NE_1 and CNCs@Th-EO NE_3) showed no changes in DLS parameters over time, contrary to the others (CNCs@Th-EO NE_2 and CNCs@Th-EO NE_4). In particular, the CNCs@Th-EO NE_2 sample (10% *v*/*v* oil without CaCl_2_) showed a doubling of the Z-average diameter with a slight reduction in PdI after 7–14 days, indicating that a first coalescence phenomenon had already occurred after 7 days without flocculation or sedimentation. This behaviour is in line with what is suggested by the NE observation (creaming phenomenon) and the stability values in Table 3. Furthermore, the formulations with the lowest oil content (CNCs@Th-EO NE_3 and CNCs@Th-EO NE_4) formed smaller nanodroplets than the 10% *v*/*v* concentrated samples. This can be explained by the increased interfacial area of the droplets due to the higher oil concentration. As the number of CNCs was the same for all formulations, there was less cellulose available to stabilise the interfacial area. Consequently, an increase in EO concentration would result in less extensive coverage of the droplets’ surface by the available CNCs, leading to an increase in droplet volume with a consequent increase in Z-average diameter that reduces the total interfacial area. Huaping Yu et al. reported that the size of the clove oil droplets produced via homogenisation depended on the contribution of the stabiliser when a low number of CNCs was used [34]. These data are in line with our outcomes for the last NEs. The same trend was observed for cellulose-based EO MEs [33,35]. Finally, we can deduce that the crosslinking by Ca^2+^ compensates for the inefficient coverage of the droplet surface due to excess oil, thus increasing the stability of NEs without increasing the CNCs amount. 

Concerning the ζ-potential parameter, any variations were recorded between samples and over time. We recorded ζ-potential values in the range of −25–−27 mV for all NEs, which remained unchanged after storage. Since Tween 80 is a nonionic surfactant, the negative ζ-potential is expected to be contributed by CNCs, which showed a ζ-potential of −36.7 ± 3 mV at 0.1 mg/mL in ultrapure water. A value of −25 mV provided a strong enough electrostatic repulsion to prevent droplet aggregation, stabilising the NEs. The increase in the ζ-potential values of NEs suggests structural changes in oil nanoparticles that may be related to instability problems, as indicated by the increase in droplet size with ageing time [60]. This destabilisation process related to NEs is known as Ostwald ripening, and it also involves a diffusive transfer of the EO from smaller to larger droplets [61]. For this reason, it is important to cover and stabilise nanodroplets with polymers or nanoparticles. In our experiments, the ζ-potential did not change over time, which is in line with the recorded slight variations in the Z-average diameter. Appendix A presents the DLS parameters of the CNCs@Th-EO NE_1 sample over time. This formulation showed excellent dilution stability; the nanoparticle size and ζ-potential remained constant even after a 1/1000 dilution (necessary to perform appropriate DLS measurements). These data are consistent with the absence of observed phase separation, as discussed in the previous section. The Z-average diameter remained around 230 nm, suggesting that the high-energy homogenisation method resulted in the spontaneous formation of CNCs-stabilised nanoscale droplets, as indicated by the recorded colloidal stability (see ζ-potential and PdI parameters). CNCs nanodimensions ensure a robust coating of OE micelles that promotes emulsification by blocking the Ostwald’s maturation mechanism [62]. An alternative way to kinetically improve EO emulsion stability is the exploitation of lipids [63]. Spherical cellulose nanocrystals produced Pickering emulsions, recording a decrease in microdroplet size and an increase in emulsion ratio as the CNC concentration increased [29]. Therefore, the morphology of the particles is also critical in producing stable emulsions and determining the size of the droplets.

Our data demonstrated that the addition of the crosslinker improves the emulsification potential of CNCs by producing a uniform distribution of emulsion nanodroplets even at the highest Th-EO concentration of 10% *v*/*v*. Therefore, the CNCs@Th-EO NE_1 formulation (10% *v*/*v* of Th-EO with CaCl_2_) was selected for the following characterisations thanks to its high Th-EO concentration and retention over time and its good kinetic stability.

#### 2.2.4. CNCs and NEs Droplet Morphology

CNCs nanodimensions in 6% *w*/*v* aqueous gel from CelluloseLab were confirmed via SEM analysis. Appendix A presents a representative SEM image of CNCs in which we can observe the nanoneedle morphology as indicated by the product data sheet. 

The morphology of CNCs@Th-EO NE_1 was characterised via TEM. First, TEM analysis revealed the formation of nanomicelles of 23.8 ± 4 nm deriving from surfactant molecule interaction with Th-EO (see Appendix A). Appendix A show the presence of CNCs, such as a network in which nanomicelles were immersed. Figure 3 presents two representative TEM images of CNCs@Th-EO NE_1 nanodroplets. 

We observed spherical and homogeneous droplets in which we could visualise the single micelles. The nanodimensions of the emulsion were also confirmed in agreement with the measured hydrodynamic diameters reported in Appendix A. TEM images displayed droplets with a diameter of 109.4 ± 18 nm; hydrodynamic diameter (~230 nm) is bigger than effective colloidal diameter because of water molecule presence. 

Our observation is in line with the literature on pesticide nanoemulsions, which occur in spherical shapes or with core shell-like structures due to a few clusters of nanomicelles, formed during preparation [16]. The small droplet size could be attributed to the effective adsorption of crosslinked CNCs on oil micelles as well as to the downsizing effect of the homogenisation process, as previously indicated by DLS data [34]. 

The observed droplet aggregation is due to rearrangement of particles when dried on a copper grid for TEM analysis. The advantages of NEs over MEs are their smaller drop sizes, which provide a formulation with increased wettability, spreadability, and bioavailability. However, NEs performance in agriculture such as in other applications, could be affected by long-term instability [64]. This characteristic was evaluated using Turbiscan analysis. 

#### 2.2.5. Long-Term Stability of Nanoemulsion with Turbiscan LabExpert

The higher encapsulation performance in the CNCs@Th-EO NE_1 formulation may be contributed by the excellent emulsifying properties of CNCs following the ionic crosslinking of Ca^2+^. Optical observation as well as DLS analysis demonstrated the physical–chemical stability of this NE. Emulsions stabilised by solid colloidal nanoparticles, referred to as Pickering emulsions, were generally characterised by long-term stability [32]; we verified this issue via Turbiscan LabExpert analysis during the first 24 h after NE preparation (following o.n. solvent evaporation) and after 30 days of storage at RT. These measurements were effectuated on native samples without any dilution. The instrument software (Turbiscan Easy Soft) calculates the TSI value (see Equation (4)) that is used to predict dispersion stability [65]; the smaller TSI value corresponds to a more stable system. In this study, TSI variation over 24 h of analysis was monitored, comparing the CNCs@Th-EO NE_1 immediately after preparation and after 30 days. TSI plots are presented in Figure 4. 

The instrument software indicated that TSI values below 0.5 correspond to a not significant chance condition, TSI values in the range 0.5–1 correspond to an early stage of destabilisation, and TSI values in the range 1–3 correspond to a destabilisation phase; values above 3 indicate great destabilisation of the system. The first three phases are not visible to the naked eye. Therefore, we observed that the CNCs@Th-EO NE_1 formulation reached an early destabilisation stage after 24 h, remaining in the destabilisation phase (TSI below 3) after 30 days of storage. These data are consistent with the stability index values in Table 3. We observed no visible change in this formulation, which remained homogeneous and without sedimentation, flocculation, or creaming processes, as just described in Section 2.2.2. These qualitative observations were confirmed via Turbiscan LabExpert analysis, which showed no ripples in the Δ backscattering (BS) plot, as reported in Appendix A. 

### 2.3. Biological Properties of Th-EO and CNCs@Th-EO NE 

The antimicrobial properties of Th-EO are well known, and research has led to a better understanding of the mechanisms responsible for its behaviour. The phenolic components of the essential oil appear to play an important role in its antimicrobial activity, particularly against bacteria, as other studies have shown. Membrane disruptions followed by depolarisation, altered ion exchange, and cytoplasmic leakage are the most likely hypotheses, although multisite action cannot be excluded [66]. Indeed, thymol as itself, which was one the main constituents of the Th-EO used in this work, was positively used to inhibit *R. solanacearum* and several *Xanthomonas* species, the causal agents of bacterial wilt and leaf spot in many crops [36]. A great growth inhibition activity of this compound has been demonstrated in *X. fastidiosa* cells and has been greatly enhanced by nanoencapsulation and smart delivery [39]. Even cymene was positively assayed for its antimicrobial properties on several microorganisms, including nematodes, fungi, and food-borne bacteria, in both its natural forms (para and ortho) [67,68]. To our knowledge, this is the first report on the antimicrobial activity of an o-cymene-based product on phytopathogenic bacteria. 

Our Th-EO antimicrobial activity was first tested on Psav via incorporation in agarised KB. The colony count revealed a complete growth inhibition comparable to that of copper sulphate at the field dose when the essential oil was used at a concentration higher than 1% *v*/*v*. Even at lower concentrations (0.5% *v*/*v*), a remarkable inhibition (85%) was achieved (see Figure 5).

In terms of biocompatibility, no adverse effects were observed on the leaf development of olive plants treated with Th-EO and the CNCs used in this work (see Table 4). As a matter of fact, NBI, which was calculated using the ratio of chlorophyll and flavanol contents in leaves, gives a reliable indication of the health status of the plants: this index considers plants to be in good shape when most of the absorbed nitrogen is used for primary metabolic functions, such as chlorophyll synthesis, while plants could be subjected to abiotic or biotic stress when secondary metabolic products demand more nitrogen, lowering the NBI values [69,70]. 

Several studies have already pointed out the absence of toxicity for plants treated with CNCs, confirming our results, but when it comes to essential oils, the outcomes can be more unpredictable, since many factors are involved in the biological activity of EOs, such as temperature, humidity, and concentration [71]. However, this experiment highlighted the possibility of using an effective antibacterial amount (0.5% *v*/*v*) of Th-EOs without damaging the olive plants’ basal growth functions, such as leaf development and nitrogen metabolism.

Given the interesting preliminary results about the biological activity of Th-EO on Psav and olive plants, we moved towards the characterisation of the antimicrobial activity of the nanoemulsion. CNCs@Th-EO NE_1 was picked from among the proposed formulations due to its good stability over time, lack of sedimentation and flocculation, and high retention of Th-EO, as previously stated. We first compared the antimicrobial activity of the selected NE with that of Th-EO alone using a simple disc diffusion test. Interestingly, when the NE was used at the same concentration as the EO, a wider inhibition halo around the disc could be appreciated, as can be observed in the plot in Figure 6. 

This could be explained by considering the encapsulated form of EO instead of the free form, which could have resulted in less volatilisation or dispersion of the oil [72]. The advantages of encapsulating EO with nanomaterials have already been reported by several authors, who pointed out an increased stability and an increase in antimicrobial activity [73,74]. Furthermore, the synergistic effect of these molecules was explored and confirmed by Abbasi et al. (2023), who proposed a Th-EO and carboxymethylcellulose coating for postharvest applications [75].

Since there was a need for a quantitative determination of CNCs@Th-EO NE_1’s inhibition activity, we tested it on Psav through a microdilution method. The count of developed colonies after it was exposed to increasing concentrations of the NE in a growth-promoting broth revealed the capability of the compound to fully inhibit the bacteria starting from doses higher than 0.05% *v*/*v*, while lower doses of 0.1% *v*/*v* of NE could inhibit the bacterial growth by 90%, still in a comparable way to the inhibition displayed by copper at the field dose (see Figure 7). 

As CNCs have been shown to have unique antimicrobial mechanisms on phytopathogenic bacteria, such as inhibition of swimming motility and biofilm production, as well as induction of cell flocculation in liquid media, we hypothesised a synergistic effect of nanocrystals and Th-EO [71]. The presence of CNCs in the formulation could have resulted in the bacterial cell being more exposed to the drug, preventing it from moving freely in the medium and adhering to surfaces, and of course preventing the EO from settling or separating from the liquid phase. This behaviour was confirmed in the artificially inoculated plants, where the epiphytic survival of Psav was monitored for 14 days (see Figure 8). 

At 1 dpi, only copper showed an appreciable effect in lowering the Psav epiphytic population (1.21 CFU/cm^2^), while the NE showed similar values to the negative control treatment (2.38 and 2.37 CFU/cm^2^, respectively). This could be explained considering the great capability of the formulation to retain the encapsulated active ingredients, preventing it from being available as soon as it comes into contact with the bacterial cells. On the contrary, a few days after the application, the inhibition activity of the EO could be appreciated. Indeed, the re-isolation of the bacteria from the leaves highlighted the capability of the compound to reduce up to one log unit (1.41 CFU/cm^2^) the bacteria survival in a comparable way to that of copper sulphate after seven days from inoculation (1.48 CFU/cm^2^). The same trend was recorded at 14 dpi, since the recovered colonies from NE- and CuSO_4_-treated plants were statistically similar (1.46 and 1.31 CFU/cm^2^, respectively). Lowering the capability of phytopathogenic bacteria to survive outside the host is a valid strategy to decrease the risk of infections, since the epiphytic phase represents an important source of inoculum [76]. As a matter of fact, Psav preventive control methods heavily revolve around the use of protectants, such as cupric salts and biocontrol agents, and the observance of good agronomic practices to avoid the opening of artificial wounds which could promote the pathogen’s ingress, especially considering the bacteria’s ability to persist on the host over the seasons [77]. For these reasons, further studies are planned to better assess the capability of the NE to maintain its antimicrobial properties over time, especially in field conditions, where environmental components, such as light, temperature, air humidity, and precipitations play a fundamental role in the kinetics of agrochemicals. Given the collected data, we believe the proposed nanoencapsulation method could provide an effective way to deliver volatile and instable compounds such as essential oils, while maintaining, if not boosting, their biological properties.

## 3. Materials and Methods

### 3.1. Materials

CNCs were purchased from CelluloseLab (212-2 Garland Court, Enterprise Bldg., Fredericton, NB, Canada, E3B 5A3), in 6% *w*/*v* aqueous gel. Licofarma s.r.l. (Via Lecce 90/92, Galatina (LE), Italy; https://www.licofarma.com/, accessed on 17 May 2019) provided thyme essential oil (Th-EO) via supercritical CO_2_ extraction. The oil was stored at room temperature and used without any purification. All chemicals and organic solvents were of the highest purity commercially available from Sigma-Aldrich (Milano, Italy). The bacterial strain of *Pseudomonas savastanoi pv. savastanoi* (Psav) PvBa206 was maintained and periodically subcultured on King’s B (KB) medium Petri dishes incubated at 27 °C for 48 h [78]. One-year-old olive tree seedlings of the cultivar Leccino that were propagated by cuttings, 50 cm high, and single-stemmed were used in this work. Seedlings were propagated by cuttings, 50 cm high, and single-stemmed. Olive tree seedlings were grown in a glasshouse at 25 ± 2 °C during the day and 16 ± 2 °C during the night with a relative air humidity of 65%.

### 3.2. Identification of EO Constituents via Gas Chromatography/Mass Spectrometry

Chemical analysis of Th-EO was performed via gas chromatography/mass spectrometry (GC-MS), using the Agilent Technologies, 7820A GC system with HP-5MS column (5% phenyl-methyl-polysiloxane; 30 m × 0.25 mm i.d., 0.2 μm film thickness). Helium was used as carrier gas at a flow rate of 1 mL/min with ionisation voltage of 70 eV. The injection volume of EO was 1 μL in hexane. The injector was maintained at 250 °C. The oven temperature was maintained at 40 °C for 1 min followed by an increment of 3 °C/min, and finally held at 250 °C. Mass range scanned was 40–160 amu. The peaks of the mass spectra obtained for the compounds were compared with those from the NIST Mass Spectrometry Data Center. 

### 3.3. Preparation of CNCs-Stabilised Th-EO Nanoemulsions

CNCs@Th-EO NEs were prepared by mixing the oil phase, Th-EO and surfactant Tween 80 in ethanol, and the water phase, CNCs in distilled water with and without 3 mM of CaCl_2_. The ratio Th-EO:Tween 80 (*v*:*v*) was maintained 2:1. We investigated two Th-EO:CNCs ratios, 1:1 and 6:1 (*w*:*w*), to prepare NEs with two Th-EO concentrations (1.5 and 10 % *v*/*v*). The oil/ethanol ratio was always 1:1 (*v*:*v*). First, CNCs suspension was mixed with 3 mM of CaCl_2_ (or only water in the experiments without CaCl_2_) with continuous stirring at 500 rpm for 5 min. Then the oil phase was gradually added to the water phase with continuous stirring, and then the mixture was emulsified using an Ultra-Turrax blender model IKA T25 (IKA Werke, Staufen, Germany) at 15,000 rpm for 10 min. NEs were agitated at 300 rpm o.n. at RT under the fume hood for ethanol evaporation. Solvent removal was easily verified by quantifying the volume reduction. The obtained samples were stored at RT and 4 °C for further characterisations. The Table 5 presents the contained components of the prepared NEs. 

### 3.4. Characterisations 

#### 3.4.1. UV–Vis Spectroscopy and Oil Retention

The readings for UV–vis spectra of Th-EO and resulting NEs were recorded in the range of 250–350 nm using a Varian-Cary 500 spectrophotometer. The unknown concentration was obtained with reference to a standard curve using Th-EO standard solutions in ethanol at known concentrations in the range of 50–0.1 mg/mL, and the line was fitted using Origin software (OriginPro 2016 64bit) (Abs values at λ_max_ = 273 nm were multiplied by dilution factor). 

Oil retention over storage time was evaluated via EO quantification, which was performed using spectrophotometric analysis to record UV-Vis absorption spectra at 273 nm, and referring to the standard curve. Oil retention percentage was calculated according to the following equation:(1)$$\mathrm{Oil}\;\mathrm{retention}\;\%=\frac{\mathrm{amount}\;\mathrm{of}\;\mathrm{EO}\;\mathrm{quantified}\;\mathrm{in}\;\mathrm{NE}}{\mathrm{amount}\;\mathrm{of}\;\mathrm{EO}\;\mathrm{added}\;\mathrm{to}\;\mathrm{NE}}\times 100$$

#### 3.4.2. Total Phenolic Content

Total phenolic content was estimated spectrophotometrically according to the Folin–Ciocalteu colorimetric method. The reaction mixture was prepared by mixing 50 µL of EO or NEs (20 µg/mL of oil), 50 µL of 10% Folin–Ciocalteu’s reagent dissolved in 750 µL of distilled water, and 500 µL of 7% NaHCO_3_. The samples were incubated at RT for 90 min in the dark. The absorbance was determined at λ_max_ = 740 nm. It was calibrated against gallic acid standards (concentration range: 20–1 µg/mL) and the results were expressed as µg gallic acid equivalents (GAE)/mg EO. Data presented are average values of three measurements for each sample.

#### 3.4.3. Stability Study

Emulsion storage stability was evaluated: visual transitions from steady state to creaming and coalescence for a period of 15 days were noted in graduated tubes. Nanoemulsion stability (%) was measured in duplicates at 0, 1, 7, and 14 days. The emulsion stability index (SI) was expressed as the formed emulsion layer volume (V_emuls_) relative to the total sample volume (V_total_)_._ The SI was calculated using the following equation:(2)$$\mathrm{SI}\;\%=\frac{V_{\mathrm{emuls}}}{V_{\mathrm{total}}}\times 100$$

Stability towards shear was evaluated using centrifugation methodology. Samples were centrifuged at 10,000 rpm for 5 min at 20 °C in graduated tubes. The centrifugation resulted in an oily phase at the top, an emulsion phase in the middle, and an aqueous phase at the bottom. Emulsion stability (ES) was determined using the following equation:(3)$$\mathrm{ES}\;\%=\frac{\mathrm{remaining}\;\mathrm{emulsion}\;\mathrm{volume}}{\mathrm{initial}\;\mathrm{emulsion}\;\mathrm{volume}}\times 100$$

#### 3.4.4. Droplet Size and Colloidal Stability

Sample dilution was performed to study stability against phase separation based on particle size distribution, polydispersity index, and ζ-potential measurements through DLS. CNCs@Th-EO NEs were analysed with a Nano ZS90 (Malvern Instruments, Cambridge, UK) instrument. Preliminary DLS tests indicated the optimum sample concentration of 0.1 mg/mL. The potential analysis of particles was carried out via laser Doppler velocimetry (LDV). Measurements were performed at RT, in filtered distilled water (0.45 µm), at the refractive index of cellulose (1.469). The ζ-potential values are reported as the mean of 5 measurements; each of them was derived from 10 different runs to establish measurement repeatability. Particle size distribution data are reported as the mean of 3 measurements, each of them derived from 15 different runs to establish measurement repeatability. 

#### 3.4.5. CNCs and NEs Morphological Analysis

CNCs were observed with scanning electronic microscopy (SEM). A drop of the sample was placed on silicon support and dried at room temperature and then was viewed under a SEM MERLIN ZEISS, with an FEG source, at an accelerating voltage of 20 kV, using short exposure time (a few tens of seconds).

The droplet morphology of CNCs@Th-EO NE was analysed with transmission electron microscopy (TEM). A drop of NE (10 μL) was placed on a standard carbon-coated 200-mesh copper grid and left at RT o.n. The TEM micrographs were acquired by analysing the grid under a JEOL JEM 1400Plus microscope with LaB6 source at an accelerating voltage of 80 kV. 

#### 3.4.6. Stability Measurement of the Nanoemulsion Using Turbiscan LabExpert

Nanoemulsion stability was estimated using Turbiscan LabExpert, after preparation and after 30 days of storage at room temperature in the dark. This instrument has two synchronous detectors and a near-infrared light source (λ = 880 nm). The transmission detector receives the light flux transmitted (T) through the sample; the backscattering detector captures the backscattered light (BS). The reading head acquires transmissions and backscattering data either at a chosen position on the sample cell or every 40 μm, while moving along the 55 mm cell height. The recorded backscattering intensity of radiation is proportional to particle concentration. An important factor obtained with the Turbiscan instrument is related to the Turbiscan stability index (TSI). It monitors the destabilisation kinetics versus ageing time. It sums all the variations detected in the sample (size and/or concentration) at a given ageing time. The higher the TSI is, the worse the stability of the sample is. These stability index results have been used to compare the stability of many samples. The formula related to the TSI is reported in Equation (4):(4)$$\mathrm{TSI}=\sqrt{\frac{\sum_{i=1}^{n}{\left(x_{i}-x_{\mathrm{BS}}\right)}^{2}}{n-1}}$$
where x_i_ is the mean backscattering for every minute of measurement, x_BS_ is the mean x_i_, and n is the number of scans [79,80]. 

Nanoemulsions without dilution were placed in a sample bottle (20 mL) and were scanned every 10 min for 24 h at 27 °C. After the internal changes were monitored, the TSI was calculated to evaluate NE stability with Turbiscan Easy Soft and provide a unique parameter.

### 3.5. Biological Study 

#### 3.5.1. Preliminary Biological Activity of Th-EO and CNCs

Extracted Th-EO was evaluated for its antimicrobial properties on Psav by incorporating it in agarised media [81]. Briefly, an aliquot of Th-EO was added to KB medium to reach the desired concentrations (0.1, 0.5, 1, 2% *v*/*v*). Then, 100 μL of a 10^3^ CFU/mL suspension made from a fresh Psav culture was uniformly streaked onto the plates. After an incubation of 48 h at 27 °C, developed colonies were counted and the growth inhibition (expressed as percentage) was calculated as follows:(5)$$\mathrm{Growth}\;\mathrm{Inhibition}\;\%=\frac{\mathrm{Negative}\;\mathrm{control}\;\mathrm{Colonies}-\mathrm{Treatment}\;\mathrm{Colonies}}{\mathrm{Negative}\;\mathrm{control}\;\mathrm{Colonies}}\times 100$$

KB alone and KB amended with 0.3% *w*/*v* copper sulphate pentahydrate were used as negative and positive controls, respectively [43]. Each thesis consisted of three replicates. The experiment was repeated twice.

Effects of Th-EO and CNCs were studied on olive plants to preventively assess the phytotoxicity of the NEs. Olive plants (10 per thesis) were spray-treated with a 0.5% *v*/*v* suspension of Th-EO and a 0.5% *v*/*v* suspension of CNCs until runoff. At 1, 7, and 14 days post-treatment (dpt), 8 measurements per plant were taken using a leaf clipper (Dualex 4 Scientific, FORCE-A, Orsay Cedex, France) to quantify the chlorophyll and the flavonol contents, expressed as Dualex unities (DUs). The nitrogen balance index (NBI) was calculated as the ratio of the two values [82]. At 14 dpt, two leaves per plant were harvested and their area was measured using the software ImageJ (version 1.51j8) (NIH, Bethesda, MD, USA) (accessed on Windows 10) (Microsoft, Redmond, WA, USA) [83]. Water was used as control. The experiment was repeated twice.

#### 3.5.2. Antimicrobial Properties of Th-EO Nanoemulsion

The highest-performing NEs in terms of kinetic stability and Th-EO content were tested on Psav to quantify its antimicrobial activity. First, a disc diffusion test was designed to assess the differences between Th-EO alone and incorporated in CNCs in terms of the inhibition halo [84]. Sterile paper discs were placed on KB plates previously inoculated with 100 μL of a 10^6^ CFU/mL Psav suspension, then 10 µL of different suspensions (0.5, 1% *v*/*v*) was pipetted on the discs. After 48 h of incubation at 27 °C, the inhibition halo was measured. Water and copper were used as controls as previously mentioned. Each thesis consisted of five replicates. 

Afterwards, the inhibition of the selected NE was deeply investigated using a microdilution test: 20 μL of a 10^4^ CFU/mL Psav suspension was added to 80 μL of Luria–Bertani broth previously amended with the substances to reach the final concentrations of 0.05, 0.1, 0.5, and 1% *v*/*v*. After 24 h of incubation at 27 °C under continuous orbital shacking, several decimal dilutions were obtained and 100 μL from each was plated on KB. After 48 h of incubation at 27 °C, developed colonies were counted [85]. The growth inhibition percentage was calculated as described in Equation (5). Each thesis consisted of three replicates. 

The antimicrobial properties of the proposed NE were eventually tested on olive plants, via observing the effects of the nanomaterials on the epiphytic survival of the bacterium. Plants were treated as reported before, then they were spray-inoculated after 24 h with a 10^6^ CFU/mL Psav suspension, they were bagged, and the air humidity of the growth chamber was raised to 80%. At 1, 7, and 14 dpi (days post-inoculation), 10 leaves per thesis (1 leaf per plant) were collected and washed in PBS (phosphate-buffered saline) using a homogeniser (Stomacher 400 Circulator, Seward Ltd., Worthing, UK) set at 150 rpm for 4 min. Several dilutions were plated on KB agar plates, which were incubated at 27 °C for 48 h. The number of developed colonies was counted and divided by the total area of the harvested leaves to obtain the final value of Log_10_ CFU/cm^2^ [31]. In each test, water and copper were used as controls. The experiments were repeated twice. 

## 4. Conclusions

The urge to find sustainable agrochemicals as alternatives to traditional pesticides has led to the exploration of innovative antimicrobial compounds. Nanoencapsulation could represent a suitable way to enhance the biological properties of natural active ingredients, which often present limitations to final field application due to volatility, phytotoxicity, and poor miscibility. In this context, essential oils represent a valid and ecofriendly alternative to chemicals and actual biopesticides.

We produced an *o*/*w* nanoemulsion of thyme essential oil derived from supercritical CO_2_ extraction and composed of *o*-cymene and thymol, which are well-known bioactive phytochemicals. Nanoemulsion preparation was studied, exploiting the stabilisation action of CNCs and varying initial concentrations of EOs and an ionic crosslinker. The presence of CNCs in the aqueous phase of the formulations was associated with great encapsulation efficiency and stability of Th-EO, even with a high oil content. However, the best physical–chemical stability over time at room temperature was obtained in the presence of a crosslinked cellulose network, which was able to maintain 100% oil retention and a constant hydrodynamic diameter of nanodroplets. These features were demonstrated via a stability study, EO quantification assays, and DLS parameter monitoring. The nanodimensions of the produced formulation were confirmed using TEM morphological analysis. UV-vis spectroscopy and a TPC assay suggested that the bioactives of Th-EO were not altered after encapsulation and remained stably entrapped for 30 days, as confirmed via Turbiscan LabExpert analysis.

Many essential oils, despite the numerous reports on their antimicrobial and antioxidant properties, have rarely found an effective delivery method based on sustainable molecules, such as organic polymers. In this work, we have demonstrated for the first time that is possible to formulate thyme essential oil with cellulose nanocrystals crosslinked by calcium ions to obtain a very stable nanoemulsion which showed greater inhibition against the causal agent of olive knot disease than free oil under the same conditions. This interesting bioactivity was demonstrated using different biological assays, both in vitro and *in planta*, highlighting the high antimicrobial activity of our nanoemulsion without displaying any phytotoxic effect. We firmly believe that this approach could be further deepened to investigate different bioactive molecules as well as different pathosystems.

## Data Availability

All data are available in the publication.

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
