# Peer review of "Cellulose Nanocrystal-Based Emulsion of Thyme Essential Oil: Preparation and Characterisation as Sustainable Crop Protection Tool"

_molecules, 2023, doi:10.3390/molecules28237884_

Round 1

Reviewer 1 Report

Comments and Suggestions for Authors

The manuscript is great. However, certain issues need the authors' attention before acceptance. The following key points outline the necessary revisions:

1. Abstract must clearly highlight the objectives of the study, and the novelty of your work.

2. To enhance the introduction and provide visual support, it is recommended to consider the inclusion of pertinent figures, photos, maps, or diagrams.

3. The methodology is well-defined; however, the inclusion of a flowchart would improve its overall clarity and understanding.

4. “The increase in oil content does not affect the loading capacity in the emulsion as well as the addition of CaCl2.” This statement is vague as no further explanation in term of conclusion or solution is provided.

5. In general, strengthen your discussion section by carefully analyzing your results and interpreting their significance. Provide a more in-depth discussion to clarify and justify the results.

6. Begin the conclusion section with a summary of your work, emphasizing the main findings and the gap your work fills in the literature.

Comments on the Quality of English Language

English is good and understandable

Author Response

Dear,

thank you for revision work.

We have evaluated all the comments pointed out by each reviewer and we are now ready with the revised version addressing all of the issues.

Any revisions to the manuscript have been highlighted in blue print font to facilitate the review work.

Please find enclosed a point-to-point description of the responses (blue print font) to the reviewers’ comments (black print font).

Reviewer #1: The manuscript is great. However, certain issues need the authors' attention before acceptance. The following key points outline the necessary revisions:

  1. Abstract must clearly highlight the objectives of the study, and the novelty of your work.

Thank you for the suggestion, abstract section has been revised to highlight the objectives and novelty of the study.

Here the revised abstract:

“Essential oil-based pesticides, which contain antimicrobial and antioxidant molecules, have potential for use in sustainable agriculture. However, these compounds have limitations such as volatility, poor water solubility, and phytotoxicity. Nanoencapsulation, through processes like micro- and nanoemulsions, can enhance the stability and bioactivity of essential oils. In this study, thyme essential oil from supercritical carbon dioxide extraction, was selected as sustainable antimicrobial tool and nanoencapsulated in an oil-in-water emulsion system. The investigated protocol provided the high-speed homogenisation in presence of cellulose nanocrystals as stabilisers, and calcium chloride as ionic crosslinking agent. Thyme essential oil was characterized by GC-MS and UV-vis analysis indicating the rich content in phenols. Cellulose nanocrystals/essential oil ratio and calcium chloride concentration were varied to tune nanoemulsions physical-chemical stability that was investigated by UV-vis, direct observation, Dynamic Light Scattering and Turbiscan analysis. Transmission electron microscopy confirmed the nanosized droplets formation. The nanoemulsion resulting from the addition of crosslinked nanocrystals was very stable over time at room temperature. It was evaluated for the first time on Pseudomonas savastanoi pv. savastanoi, the causal agent of olive knot disease. In vitro tests showed a synergistic effect of the formulation components, and in vivo tests on olive seedlings demonstrated reduced bacterial colonies without any phytotoxic effect. These findings suggest that crosslinked cellulose nanocrystals-emulsion can enhance the stability and bioactivity of thyme essential oil, providing a new tool for crop protection”.

  1. To enhance the introduction and provide visual support, it is recommended to consider the inclusion of pertinent figures, photos, maps, or diagrams.

Thank you for the suggestion, the scheme 1 has been added in the introduction section to represent the main properties of nanoemulsions in agriculture, in particular for essential oils application.

  1. The methodology is well-defined; however, the inclusion of a flowchart would improve its overall clarity and understanding.

Thank you for the suggestion, the scheme 2 has been added in 2.2.1 paragraph to provide a flowchart of methodologies.

  1. “The increase in oil content does not affect the loading capacity in the emulsion as well as the addition of CaCl2.” This statement is vague as no further explanation in term of conclusion or solution is provided.

Thank you for the suggestion, this statement has been revised to better clarify and explain the data (see lines 205-209).

  1. In general, strengthen your discussion section by carefully analyzing your results and interpreting their significance. Provide a more in-depth discussion to clarify and justify the results.

Thank you for the suggestion, some results have been deeply discussed. The added sections have been highlighted in blue print font to facilitate the review work.

  1. Begin the conclusion section with a summary of your work, emphasizing the main findings and the gap your work fills in the literature.

Thank you for the suggestion, conclusion section has been revised.

Here the revised conclusion:

“The urge to find alternative and sustainable agrochemicals to traditional pesticides, has led to the exploration of innovative antimicrobial compounds. Nanoencapsulation could represent a suitable way to enhance the biological properties of natural active ingredients, which often present limitations to field final application due to volatility, phytotoxicity and poor miscibility. In this context, essential oils represent a valid and eco-friendly alternative to chemicals and actual biopesticides.

We have produced a o/w nanoemulsion of thyme essential oil deriving from supercritical CO2 extraction and resulting composed in o-Cymene and Thymol, which are well known bioactive phytochemicals. Nanoemulsion preparation was studied exploiting stabilisation action of CNCs and varying initial concentration of EO and ionic crosslinker. The presence of CNCs in the aqueous phase of the formulations was associated with great encapsulation efficiency and stability of Th-EO, even at high oil content. However, the best physical-chemical stability over time at room temperature was obtained in presence of a crosslinked cellulose network, which was able to maintain 100% oil retention and a constant hydrodynamic diameter of nanodroplets. These features have been demonstrated by stability study, EO quantification assays and DLS parameters monitoring. The nanodimensions of the produced formulation was confirmed by TEM morphological analysis. UV-vis spectroscopy and the TPC assay suggested that the bioactives of Th-EO were not altered after encapsulation and remained stably entrapped for 30 days, as confirmed by Turbiscan LabExpert analysis.

Many essential oils, despite the numerous reports on their antimicrobial and antioxidant properties, have rarely found an effective delivery method based on sustainable molecules, such as organic polymers. In this work, we have demonstrated for the first time that is possible to formulate thyme essential oil with cellulose nanocrystals crosslinked by calcium ions to obtain a very stable nanoemulsion, that showed greater inhibition against the causal agent of olive knot disease than free oil under the same conditions. This interesting bioactivity was demonstrated by different biological assays, both in vitro and in planta, highlighting the high antimicrobial activity of our nanoemulsion without any phytotoxic effect. We firmly believe that this approach could be further deepened to investigate different bioactive molecules as well as different pathosystems”.

Comments on the Quality of English Language

English is good and understandable.

Thank you for the comment.

 We hope you find the manuscript in order and request you consider for publication.

Yours Sincerely,

Francesca Baldassarre

Reviewer 2 Report

Comments and Suggestions for Authors

The article by Baldassarre F. et al. describes the preparation of oil-in-water nanoemulsions containing thyme essential oil and stabilized with a combination of nanocellulose, Tween 80, and calcium chloride. The authors prepared emulsions of different formulations and evaluated their stability, morphology, and ability to resist harmful microorganisms without injury to plants. The article is well-written, deals with a relevant topic, and can be published after some corrections.

Specific comments are as follows.

Line 86: “Cellulose nanocrystals (CNCs), cellulose nanofibers (CNFs) or microfibrillated cellulose 86 (MFC), have been used in NEs and MEs, both in native and modified forms [26–28].” The authors cite articles that were published ten years ago, whereas there are newer papers, e.g., 10.1016/j.foodchem.2023.137597, 10.1016/j.carbpol.2023.120896, 10.1016/j.carbpol.2023.121429.

Line 118: “CaCl2 for a better emulsification efficiency”. It is unclear from the introduction what this effectiveness is. In the abstract, the authors write "calcium chloride was added as a crosslinking agent", but there is no mention of this in the introduction. In addition, a scheme of the action of calcium chloride as a crosslinking agent would be helpful. Probably, the crosslinking is due to the presence of sulfonic groups on the surface of CNC particles.

Line 119: “visive method”. Visible method?

Line 171: “The oil/surfactant ratio”. It is necessary to specify whether this is a mass or volume ratio.

Table 2. The data must be rounded. For example, it makes no sense to include tenths in the standard deviation in "236±11.6". It should be "236±12." The same needs to be done in the Supplementary Material and everywhere else.

Line 211: “after solvent evaporation”. This step is unclear. Additional explanations are needed.

Line 504: “We have investigate two 504 Th-EO:CNCs ratios,1:1 and 6:1 (mg:mg)…” The description of the formulations is hard to follow. It would be good if the authors would give a table containing all components of emulsions and their concentration in the resulting emulsions.

Line 511: “at RT for ethanol evaporation”. It is very doubtful that ethanol flew away under these conditions. Did the authors experimentally verify this claim somehow?

Comments on the Quality of English Language

The English language needs some editing.

Author Response

(The authors gave the same response as above.)
